# Symmetrical arrangement of positively charged residues around the 5-fold axes of SAT type foot-and-mouth disease virus enhances cell culture of field viruses

**Melanie Chitray**[1,2☯], **Abhay Kotecha**[3☯], **Peninah Nsamba**[1,4,5], **Jingshan Ren**[3], **Sonja Maree**[1], **Tovhowani Ramulongo**[1,2], **Guntram Paul**[6], **Jacques Theron**[2], **Elizabeth E. Fry**[3], **David I. Stuart**[3], **Francois F. Maree**[1,2]*

**1** Vaccine and Diagnostic Development Programme, Onderstepoort Veterinary Institute, Agricultural Research Council, Onderstepoort, South Africa, **2** Department of Biochemistry, Genetics and Microbiology, Faculty of Natural and Agricultural Sciences, University of Pretoria, Pretoria, South Africa, **3** Division of Structural Biology, The Henry Wellcome Building for Genomic Medicine, Roosevelt Drive, Headington, Oxford, United Kingdom, **4** Department of Veterinary Tropical Diseases, Faculty of Veterinary Sciences, University of Pretoria, Onderstepoort, South Africa, **5** Makerere University, College of Veterinary Medicine, Animal Resources and Biosecurity, Kampala, Uganda, **6** MSD Animal Health, Cologne, Germany

☯ These authors contributed equally to this work.
* mareef@arc.agric.za

**Data Availability Statement:** All relevant data are within the manuscript and its Supporting Information files.

## Abstract

Field isolates of foot-and-mouth disease viruses (FMDVs) utilize integrin-mediated cell entry but many, including Southern African Territories (SAT) viruses, are difficult to adapt to BHK-21 cells, thus hampering large-scale propagation of vaccine antigen. However, FMDVs acquire the ability to bind to cell surface heparan sulphate proteoglycans, following serial cytolytic infections in cell culture, likely by the selection of rapidly replicating FMDV variants. In this study, fourteen SAT1 and SAT2 viruses, serially passaged in BHK-21 cells, were virulent in CHO-K1 cells and displayed enhanced affinity for heparan, as opposed to their low-passage counterparts. Comparative sequence analysis revealed the fixation of positively charged residues clustered close to the icosahedral 5-fold axes of the virus, at amino acid positions 83–85 in the βD-βE loop and 110–112 in the βF-βG loop of VP1 upon adaptation to cultured cells. Molecular docking simulations confirmed enhanced binding of heparan sulphate to a model of the adapted SAT1 virus, with the region around VP1 arginine 112 contributing the most to binding. Using this information, eight chimeric field strain mutant viruses were constructed with additional positive charges in repeated clusters on the virion surface. Five of these bound heparan sulphate with expanded cell tropism, which should facilitate large-scale propagation. However, only positively charged residues at position 110–112 of VP1 enhanced infectivity of BHK-21 cells. The symmetrical arrangement of even a single amino acid residue in the FMD virion is a powerful strategy enabling the virus to generate novel receptor binding and alternative host-cell interactions.

**Funding:** This work was supported by funding from MSD Animal Health (previously Intervet) and the Wellcome Trust Translation Award (grant 089755). Dr P. Nsamba was a recipient of a fellowship from the Organization for Women in Science for the Developing World (OWSD) from 2007-2010 and Dr A. Kotecha was funded by the Wellcome Trust. Dr M. Chitray received a NRF UK/SA Researchers Link travel grant (UK160225159174). The work of the WT Centre in Oxford is supported by the WT core award 090532/Z/09/Z. Drs D.I. Stuart and E.E. Fry are supported by the UK MRC (grant nos. G100099 and MR/N00065X/1 to D.I.S.). The funders had no role in study design, data collection and analysis, decision to publish, or preparation of the manuscript.

**Competing interests:** The authors have declared that no competing interests exist.

## Author summary

Foot-and-mouth disease (FMD) is a major threat for world economies. Local farmers in endemic areas relying on livestock for their livelihoods are affected by FMD outbreaks. Improved vaccines are essential for effective control and should be produced based on currently circulating field viruses. Current vaccines are produced by chemical inactivation of virus grown in large-scale production cultures of BHK-21 cells. However, not all field viruses can be adapted to cell cultures. The significance of our research is in identifying amino acid residues responsible for enhancing cell culture adaptation of FMD field viruses. This study engineered chimeric viruses containing the antigenic region of a field strain and novel cell culture receptor binding sites. This facilitated rapid virus amplification within a few passages in BHK-21 cells to create master virus seed stocks, circumventing the need for isolation on primary cell lines prior to further adaptation.

## Introduction

Foot-and-mouth disease virus (FMDV) is a small, non-enveloped, icosahedral virus with a polyadenylated, single-stranded, positive-sense RNA genome belonging to the *Aphthovirus* genus in the *Picornaviridae* family [1]. The virus capsid comprises 60 copies each of four virus-encoded structural proteins, VP1 to VP4. The outer shell of the capsid contains VP1, VP2 and VP3, whilst VP4 lines the interior surface [2]. FMDV is an important pathogen that causes a highly contagious, vesicular disease affecting cloven-hoofed animals, including cattle, pigs, goats, sheep and buffalo, with severe economic consequences worldwide [3].

FMDV naturally infects epithelial cells and to date studies have shown that FMDV can adhere to any of four members of the $\alpha_V$ subgroup of the integrin family of cellular receptors, *i.e.* $\alpha_V\beta1$, $\alpha_V\beta3$, $\alpha_V\beta6$ and $\alpha_V\beta8$ [4–11]. Attachment to these receptors is mediated via a highly conserved Arg-Gly-Asp (RGD) motif [12–15], located within the structurally disordered $\beta$G-$\beta$H loop of VP1 [2,16,17]. Following FMDV-receptor interactions, the virus is internalized and the viral genome is released into the cytosol via acid-induced capsid dissociation [18,19].

Although FMDV infection is mediated by the RGD motif, RGD-independent infection can also occur [20–25]. Molecules such as cell-surface glycosaminoglycans (GAGs) have been implicated in FMDV infections in cultured cells and may be involved in RGD-independent infection [21,26,27]. Heparan sulphate proteoglycan (HSPG) is an example of such a GAG. The interactions between FMDV variants and heparin is known at atomic resolution for two strains of the virus, but the molecular basis of this interaction and mechanism of cell entry by other strains are not well understood [28,29]. O'Donnell *et al.* [30] reported that FMDV interacts with GAGs and enters cultured cells via caveola-mediated endocytosis. Recently, Lawrence *et al.* [25] demonstrated that FMDV can utilize the Jumonji C-domain containing protein (JMJD6) as part of an RGD-independent infection pathway.

It has been documented that variants of FMDV, virulent to cultured cells, emerge following serial cytolytic infections [31–33]. By comparison to parental field virus, these culture-adapted, virulent FMDV strains displayed a shorter replication cycle in BHK-21 cells and an enhanced ability to kill cells [32]. It is thought that FMDV adaptation to cell culture is made possible by selective pressure on the viral quasi-species, exerted by cell surface molecules that may act as virus receptors [20]. Cell culture growth often selects for viruses that bind to and initiate infection through cell-surface sulphated GAGs. The genetic alterations associated with increased cell tropism following cytolytic passages of FMDV in BHK-21 cells have been mapped to surface-exposed loops in the VP3 and VP1 proteins of serotype A and O viruses

[21,24,26,27,34,35]. However, it has been noted that field Southern African Territories (SAT) viruses are difficult to adapt to BHK-21 cells, thus hampering large-scale propagation of vaccine antigen [36].

Most information regarding sulphated GAG binding has come from studies with serotype O FMDV. A positive charge cluster that binds to heparin sulphate (HS) was identified within a shallow depression in the center of the biological protomer. This depression accommodates four or five heparin residues to make multiple contacts with all three outer capsid proteins [27,28,35]. Although this local structure is conserved in field viruses, one residue substitution from His (H) to Arg (R) in the VP3 protein was responsible for the acquisition to bind to HS through ionic interaction and the cell culture-adapted phenotype. In serotype C FMDV, the residues in the capsid proteins that contribute to HS binding appear to be different from those of serotype O although residue 173 of VP3 is strongly implicated to play a major role [20,37]. Cell culture adapted serotype A and SAT1 viruses have also been reported to use cell receptors other than integrins to gain entry into cells. A symmetrical amino acid cluster around the 5-fold axis of the virion has been implicated in cell culture adaptation [29,34,36,38]. The introduction of positive charge residues at position 110–111 of the VP1 protein of SAT1 viruses was critical for its extended cell tropism and cell entry [36]. However, the interaction of these residues with cellular receptors is unknown. The selection of residues following cell culture adaptation seems to be more variable in position for serotype A FMDV. A HS-binding site structurally similar to serotype O was identified for FMDV A10/Argentina/61 ($A_{10}$61), which included R residues at position 56 of VP3 and 135 of VP2 [29]. Rieder *et al*. [22] found that cell culture adaption of FMDV A12 selected for variants with residue changes near the RGD motif in the VP1 protein while Berryman *et al*. [34] reported a Gln (Q) to Lys (K) substitution in VP1-110 of a cell culture-adapted A/Turkey/2/2006 virus. FMDV A/IRN/87 had capsid residue changes on the VP2 protein that accounted for extended cell tropism of a variant of the virus [24]. For SAT2 and the many subtypes of SAT1 viruses, the mechanism of cell culture adaption is less clear; we have found positive charge residues in the VP1 protein (positions 83 and 161) that are arranged as symmetrical clusters on the capsid, but the introduction of these residues in an infectious genome-length clone did not provide growth advantage in BHK-21 cells. The molecular basis of the interaction of SAT viruses with HS and the mechanism of cell entry in cultured cells is still elusive.

Here, we provide the structural basis for cell culture adaptation of SAT1 and SAT2 viruses and showed the interaction of the capsids with HS to allow the effective entry into cultured cells. We mapped novel amino acid substitutions within the VP1 and VP3 capsid proteins of SAT1 and SAT2 viruses that are consistent with a binding site, at a position close to the icosahedral 5-fold axis, for a moiety of roughly the size and charge of a sulphated GAG. Utilizing genetically engineered variants of SAT1 and SAT2 viruses, we investigated the effect of these mutations on infectivity, cell binding and heparan sulphate (HS) dependence in cultured cells. The results of this study suggest that five of these mutant viruses have a specific affinity for HS with expanded cell tropism, which should facilitate large-scale propagation. However, only positively charged residues at position 110–112 of VP1 enhanced infectivity of BHK-21 cells through the interaction with cell-surface HSPG.

## Results

### Multiple serial passages of SAT viruses in cultured cells select for variants with increased virulence in BHK-21 cells

SAT serotypes of FMDV exhibit altered viral properties after serial passaging in BHK-21 cells [38]. To further investigate this, we serially passaged fourteen SAT1 and SAT2 viruses eight

times on BHK-21 cells (Table 1). Plaque morphologies changed from medium (3–5 mm) or large (6–8 mm) opaque plaques (parental virus isolates) to a diverse population of large or medium and small (1–3 mm) clear plaques. Complete CPE occurred within 24 h, as opposed to 48 h for the parental viruses, and the maximum titres observed were at least 10-fold higher than that of the parental viruses. The viruses varied in their ability to adapt to BHK-21 cells. However, a feature consistent with adaptation was the appearance of small plaques on BHK-21 cells with the exception of SAT2/SAU/6/00 (Table 1). The increased virulence in BHK-21 cells was associated with the ability to infect and replicate in CHO-K1 cells that lack the integrin receptors known to be utilized by FMDV for cell entry, but has the alternative GAG receptors. Additionally, the BHK-21-adapted SAT1 and SAT2 viruses, except for SAT2/UGA/2/02 and SAT1/UGA/1/97, did not replicate in CHO-677 (HS⁻) or CHO-745 [HS⁻, chondroitin sulphate deficient (CS⁻)] cells. The titres of SAT2/UGA/2/02 and SAT1/UGA/1/97 on CHO-745 cells were in the order of $10^3$ PFU/ml and $10^4$ PFU/ml, respectively (Table 1). All viruses showed plaque formation on CHO-Lec2 cells (SA⁻), demonstrating that cell entry of the SAT viruses is independent of sialic acid.

For reasons not well understood, many FMD viruses do not readily select for small, clear-plaque variants upon passage in cultured BHK-21 cells. One SAT2 isolate, SAU/6/00 calf

**Table 1. Cytolytic passaging of the SAT1 and SAT2 viruses in BHK-21 cells: the plaque morphologies and their titres in PFU/ml on BHK-21, CHO-K1, CHO-677, CHO-745 and CHO-Lec2 cells are indicated.**

| Viruses | Cell lines and plaque morphology | | | | | | | | |
|---|---|---|---|---|---|---|---|---|---|
| | Low Passage Isolates§ | | BHK-21 Adapted Isolates* | | | | | | |
| SAT1 | BHK-21 | Plaque morphology# | BHK-21 | Plaque morphology | + Heparin (1mg/ml) %reduction in number of plaques¥ | CHO-K1 | CHO-677 | CHO-745 | CHO-Lec2 |
| KNP/148/91 | $4.5×10^4$ | L, opaque | $1.2×10^8$ | M, S, clear | 89 | $9×10^7$ | 0 | 0 | $3.9×10^6$ |
| KNP/41/95 | $8.1×10^4$ | M, L, opaque | $2.7×10^7$ | L, S, clear | 80 | $4.5××10^6$ | 0 | 0 | $1.4×10^6$ |
| ZIM/13/90 | $1.8×10^3$ | M, opaque | $3.0×10^5$ | L, S, clear | 91 | $1.5×10^5$ | 0 | 0 | $6.5×10^4$ |
| KEN/5/98 | $5.4×10^3$ | M, L, opaque | $9.4×10^5$ | L, S, clear | 93 | $1.4×10^5$ | 0 | 0 | $2.3×10^3$ |
| TAN/1/99 | $5.0×10^3$ | L, opaque | $1.8×10^7$ | S, clear | 95 | $7.6×10^5$ | 0 | 0 | $2.9×10^4$ |
| UGA/1/97 | $4.5×10^2$ | M, opaque | $1.0×10^7$ | M, S, clear | 92 | $1.7×10^6$ | $3.5×10^5$ | $6.4×10^3$ | $1.3×10^5$ |
| SUD/3/76 | $6.6×10^4$ | L, opaque | $4.4×10^7$ | L, S, clear | 97 | $3.4×10^6$ | 0 | 0 | $8.0×10^6$ |
| NIG/5/81 | $4.0×10^3$ | M, opaque | $7.2×10^6$ | L, S, clear | 78 | $7.2×10^5$ | 0 | 0 | $1.6×10^4$ |
| NIG/15/75 | $3.5×10^2$ | M, opaque | $2.8×10^7$ | L, S, clear | 82 | $1.8×10^6$ | 0 | 0 | $3.5×10^5$ |
| NIG/6/76 | $1.6×10^2$ | M, opaque | $6.8×10^6$ | L, S, clear | 98 | $8.6×10^5$ | 0 | 0 | $1.2×10^5$ |
| **SAT2** | | | | | | | | | |
| KNP/2/89 | $1.0×10^5$ | M, L, opaque | $1.0×10^7$ | L, S, clear | 96 | $1.4×10^6$ | 0 | 0 | $1.7×10^4$ |
| KNP/51/93 | $2.9×10^3$ | L, opaque | $8.6×10^5$ | L, S, clear | 96 | $5.8×10^7$ | 0 | 0 | $6.1×10^4$ |
| ZIM/10/91 | $3.8×10^2$ | L, opaque | $2.0×10^7$ | L, S, clear | 96 | $1.4×10^6$ | 0 | 0 | $5.6×10^4$ |
| UGA/2/02 | $2.3×10^2$ | L, opaque | $1.4×10^7$ | M, S, clear | 69 | $2.1×10^6$ | $1.6×10^5$ | $1.0×10^4$ | $6.6×10^5$ |
| SAU/6/00 | $1.0×10^5$ | M, opaque | $1.6×10^5$ | L, clear | 0 | 0 | 0 | 0 | 0 |

§ The passage histories have been described in Maree et al. [38]. None of the low passage viruses grew on CHO-K1 cells.

# The plaque morphologies prior to BHK-21 cytolytic passage are depicted as L for large plaques, M for medium plaques and S for small plaques. Prior to cytolytic passage. Large plaques were defined as those with a diameter of 6–8 mm; medium plaques were 3–5 mm in diameter. Plaque sizes are based on three repeats of titrations and average measure of 30 plaques.

* Subsequent to cytolytic passage. Small plaques had a diameter of 1–3 mm.

¥ The average % reduction in the number of plaques on BHK-21 cells in the presence of 1mg/ml heparin from two experiments.

thyroid (CT) passage of one (SAT2/SAU/6/00$^{CT1}$), which had been responsible for a severe outbreak in dairy herds in Saudi Arabia in 2000, was passaged 58 times in BHK-21 suspension cells (SAU/6/00$^{BHK58}$) and then subjected to plaque assays. The plaque morphology of SAT2/SAU/6/00 on BHK-21 cells was medium-sized opaque plaques, whereas SAU$^{BHK58}$ produced large, clear plaques and complete CPE at 24 hours post-infection (hpi). The opaque plaque virus in the SAU$^{BHK58}$ population was undetectable. However, neither SAT2/SAU/6/00$^{CT1}$ nor SAU/6/00$^{BHK58}$ were able to infect and replicate in CHO-K1 cells or its derivatives (Table 1).

The BHK-21-adapted SAT1 and SAT2 viruses were investigated to determine if sulphated compounds could reduce infection of the SAT viruses that attained the ability to infect CHO-K1 cells. Research has shown that heparin, a soluble GAG structurally closely related to HS, can interact with some viruses and inhibit virus growth [28,38,39]. Plaque titrations performed in the presence of heparin (1 mg/ml) or absence of heparin indicated that heparin reduced plaque formation by 69–98% in BHK-21 cells (Table 1). In CHO-K1 cells, the SAT1- and SAT2-adapted viruses were inhibited by 0.625 mg/ml heparin, with the exception of SAT2/UGA/2/02 and SAT1/UGA/1/97, which were able to produce plaques in the presence of 10 mg/ml heparin (Table 1). The results indicate that SAT1 and SAT2 viruses with a broader cell tropism than field isolates can infect cells independently of the known integrin receptors for FMDV. For the majority of the viruses in this study, the results suggest that cell entry is most likely mediated by cell surface HSPG molecules.

## Gain of net positive charge in the VP1 and VP3 proteins during growth in BHK-21 cells

Analysis of the capsid proteins of the serially passaged SAT1 and SAT2 viruses revealed that the inner VP4 protein remained unchanged in all the viruses. In contrast, each virus had generally acquired three to five amino acid substitutions within VP1, VP2 and VP3 (S1 Table). Exceptions included the SAT1 virus, KNP/148/91, which showed 13 amino acid residue substitutions in the outer capsid proteins, while the SAT2 virus, UGA/2/02 had seven amino acid substitutions (S1 Table). A total of 64 amino acid residue substitutions were thus observed amongst the fifteen serially passaged viruses of which 39% ($n = 25$) were to positively charged residues (Table 2). At least eight SAT1 viruses contained K or R residue substitutions at VP1 residue positions 111 and/or 112 (βF-βG loop), and five SAT1 and SAT2 viruses had similar substitutions at VP1 positions 83, 84 or 85 (Table 2).

SAU$^{BHK58}$ did not replicate in CHO-K1 cells. Comparative amino acid sequence analysis of the outer capsid proteins of SAU$^{BHK58}$ and SAT2/SAU/6/00 revealed five amino acid substitutions, including T99A, located in the βC-βD loop of VP2, D193N in VP3, and three substitutions in VP1, *i.e.* V50L, D55N (βB-βC loop of VP1) and T158K (βG-βH loop of VP1) (S1 Table). One substitution, residue 158 of VP1, was to a positively charged lysine (Table 2).

When plotted on the 3D structure of a SAT1 capsid (2WZR) and a SAT2 capsid (5ACA) using the RIVEM program [40], most of the mutations were surface-exposed and located on protruding structural elements surrounding the five- and three-fold axes of the virion (Fig 1A and 1B). The VP1 βF-βG loop substitutions, inducing a positive charge amino acid change at positions 111 and 112, were observed in eight of the 15 viruses, are highly surface-exposed and adjacent to the 5-fold pore of the capsid (Fig 1B). Another mutation that occurred more than once in SAT1 ($n = 2$) was a lysine residue at VP1 position 84 in the βD-βE loop. Five copies of positively charged residues at both positions (111–112 and 84) in the VP1 protein formed a tight cluster on the SAT1 capsid around the 5-fold axis of symmetry (Fig 1B). For the SAT2 serotype, lysine residues were observed in VP1 at position 83 in two viruses and an arginine substitution in VP1 at position 85 for SAT2/KNP/2/89. In the current model of the SAT2

Table 2. Summary of the amino acid substitutions causing a change in the charge in the outer capsid proteins of serially passaged SAT1 and SAT2 viruses.

| Viruses | Capsid Protein | Amino acid position and substitution |
|---|---|---|
| SAT1 | | |
| KNP/148/91 | VP2 | Q74R |
| | VP1 | G112R |
| KNP/41/95 | VP1 | E84K, N111K |
| ZIM/13/90 | VP1 | N111K, G112R |
| KEN/5/98 | VP1 | E84K, W87R |
| TAN/1/99 | VP1 | E112K, D181N |
| UGA/1/97 | VP2 | E133K |
| | VP1 | E58K |
| SUD/3/76 | VP1 | C26R, N111K, G112R |
| NIG/5/81 | VP1 | N48K, N111K |
| NIG/15/75 | VP1 | N111K, D180N |
| NIG/6/76 | VP1 | N111K |
| NAM/307/98* | VP3 | E135K, E175K |
| SAT2 | | |
| KNP/2/89 | VP1 | D83N Q85R |
| KNP/51/93 | VP1 | E83K, D110G |
| UGA/2/02 | VP1 | E83K |
| SAU/6/00 | VP1 | V50L, D55N, T158K |

*FMDV SAT1/NAM/307/98 amino acid substitutions were published previously in Maree *et al.* [38].

capsid, residue 83 is not surface-exposed but amino acid 85 is exposed, forming a positively charged cluster around the 5-fold axis (Fig 1C). The positively charged cluster around the 5-fold axis would likely permit binding to negatively charged sulphated proteoglycan molecules at the cell surface.

To gain insight into how the positively charged substitutions at VP1 positions 111 and 112 in SAT1 may exert an effect on the interaction with HSPG, we used the program GRID [41] to find the most energetically favorable binding site. This procedure calculates the interaction energy of specific simple chemical probes (in this case a sulphate) at a grid of possible interaction points around a known structure. These calculations identified the most likely residue to interact with heparin as residue 112 of VP1, with a molecular interaction energy of -8.2 kcal/mol (Fig 2A and 2B). The interaction energy increased to -10 kcal/mol when the grid was centered at residue 112 (Fig 2B). Led by this result, a pentamer of heparin disaccharide units [L-iduronic acid (Idu) and D-glucosamine (GlcN)] was docked (see Materials and Methods) to both the wild-type capsid (non-substituted) and the modelled cell-adapted mutant capsid, displaying a positively charged cluster at the 5-fold axis. Fig 2 suggests that in the vicinity of the 5-fold axis, a heparin oligosaccharide can dock efficiently to the capsid containing the positively charged cluster.

Collectively, the accumulated positively charged residues around the 5-fold axis are especially notable as those alone correlated with the observed phenotypes of viruses capable of infecting CHO-K1 cells and were absent in viruses that lacked this ability.

## Generation of recombinant FMDV with altered surface charges

To study the effect of individual mutations in a defined genetic background, we selected infectious, chimeric clones of FMDV, p$^{SAU}$SAT2 (intra-serotype) and p$^{NAM}$SAT2 (SAT1-SAT2

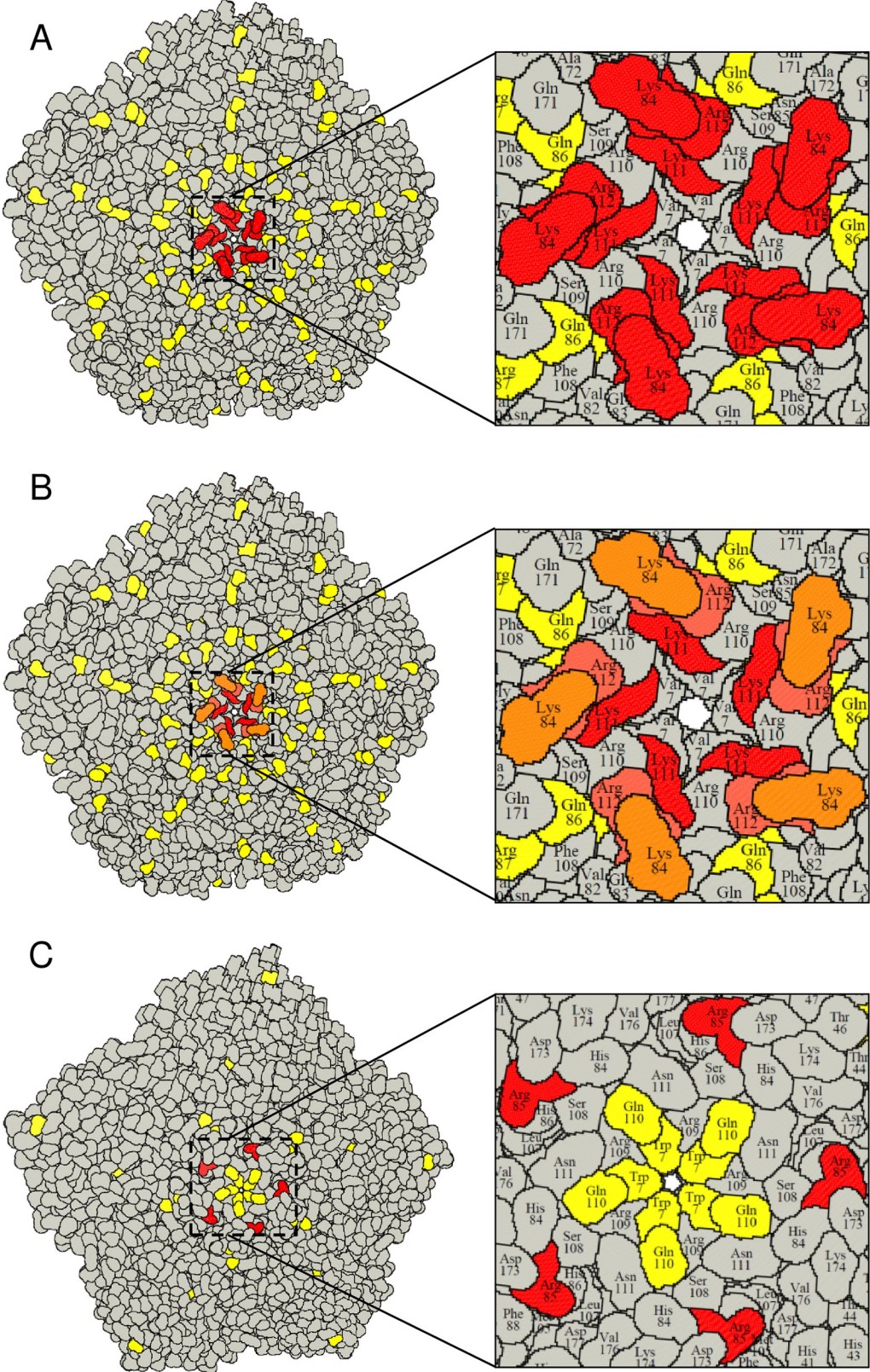

**Fig 1. A RIVEM representation [40] of the SAT1 and SAT2 pentamers. (A)** The SAT1 and SAT2 pentamers are based on the protein data bank co-ordinates 2WZR and 5ACA, respectively. Amino acid substitutions observed during the

adaptation of SAT1 viruses in BHK-21 cells are indicated in yellow. The surface-exposed, positively charged mutations that occurred more than once in different SAT1 viruses, are highlighted in red. The five copies of VP1 show the positively charged residues cluster at the 5-fold axis. (**B**) Positively charged mutations are color-coded based on the frequency of occurrence in different viruses within the SAT1 serotype from orange ($n > 1$) to red ($n > 5$). (**C**) The SAT2 pentamer is modelled using the SAT1 co-ordinates as a template and the surface-exposed, positively charged mutations are shown in red. In SAT2, a Lys residue appeared twice in VP1 position 83 in two different viruses; however, in the current model VP1 83 is not surface exposed. Nonetheless, VP1 85R (seen in SAT2/KNP/2/89) is surface-exposed.

inter-serotype), to perform the mutational studies [27,34,38]. Out of 25 different positively charged residue substitutions observed following the serial passage experiments, we selected eight patterns for the generation of recombinant viruses: (i) VP3 E135K; (ii) VP3 E135K, E175K; (iii) VP3 E158K; (iv) VP1 V50L, D55N; (v) VP1 T158K; (vi) VP1 E83K; (vii) VP1 E83K, T85R; and (viii) VP1 [110]KGG[112] to [110]KRR[112].

The effect of these mutations on the surface charge distribution of VP1 and VP3 is detailed in Fig 3. All substitutions to positively charged residues occurred near already existing positively charged surface residues, resulting in the formation of an expanded, repeat cluster of positive charge on the virion surface, especially focused around the 5-fold axis (Fig 3).

Production of progeny virus was determined upon transfection of BHK-21 cells with RNA transcripts derived from each cDNA clone. The corresponding recombinant viruses (rv), designated rv[NAM]VP3[135K], rv[NAM]VP3[135K,175K] (inter-serotype), rv[SAU]VP3[158K], rv[SAU]VP1[50L,55N], rv[SAU]VP1[83K], rv[SAU]VP1[83K,85R], rv[SAU]VP1[110KRR] and rv[SAU]VP1[158K] (intra-serotype), were recovered and high-titre stocks were prepared in BHK-21 cells and used for subsequent

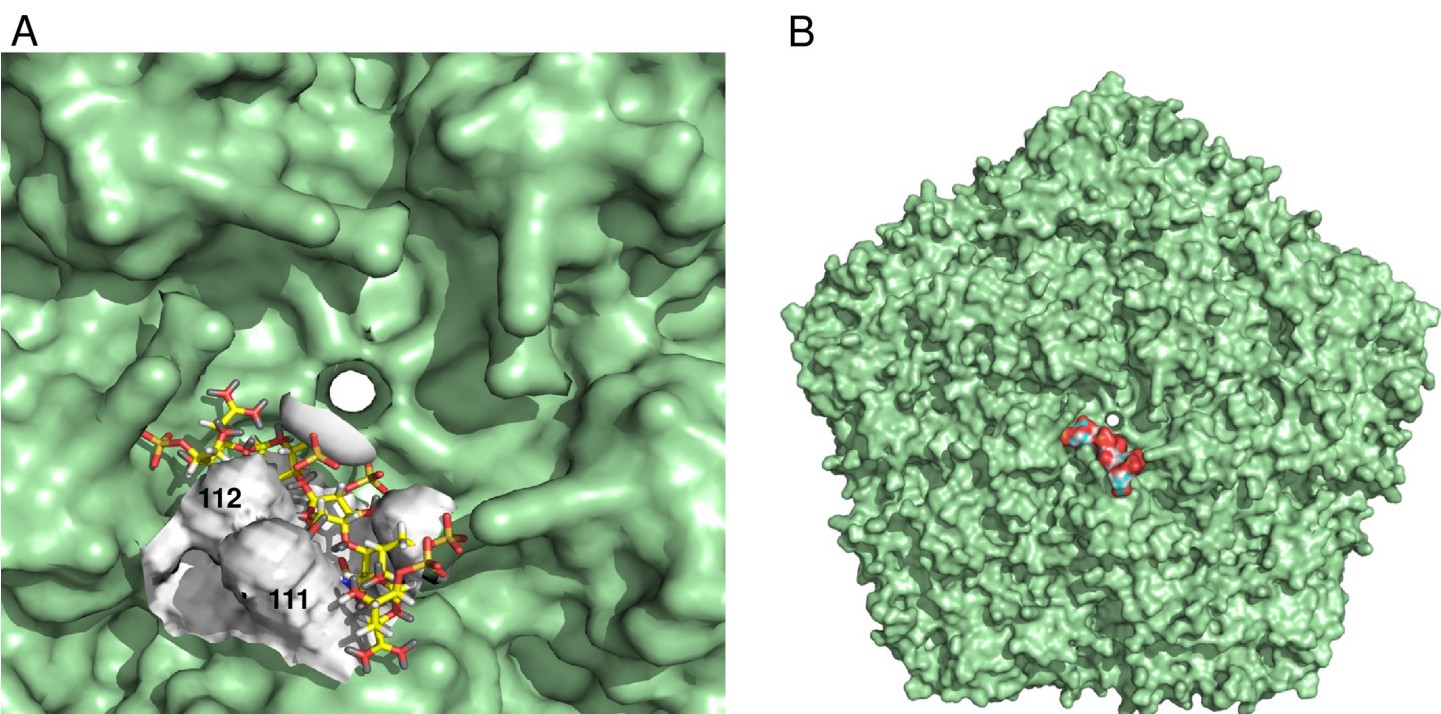

**Fig 2. GRID [41] was used to find the energetically favorable binding site for HSPG on the SAT1 modelled mutant capsid.** (**A**) The GRID calculation was performed for a 20 Å radius around the 5-fold axis using pyramidal sulfur as a probe. VP1 residue 112 is the most likely site of interaction with molecular interaction energy of -8.2 kcal/mole. The interaction energy increased to -10 kcal/mole when the grid was centered at VP1 residue 112. (**B**) Five linked heparin disaccharide molecules were docked using the default parameters of GOLD onto the SAT1 modeled mutant pentamer structures. A 30Å³ region from VP1 residue 112 was defined for docking and the GOLD fitness score function was used to rank the docking poses. The best docking pose is shown (GOLD score = 127). The equivalent process for the wild-type virus produced a less satisfactory docking (GOLD score = 102, docking not shown).

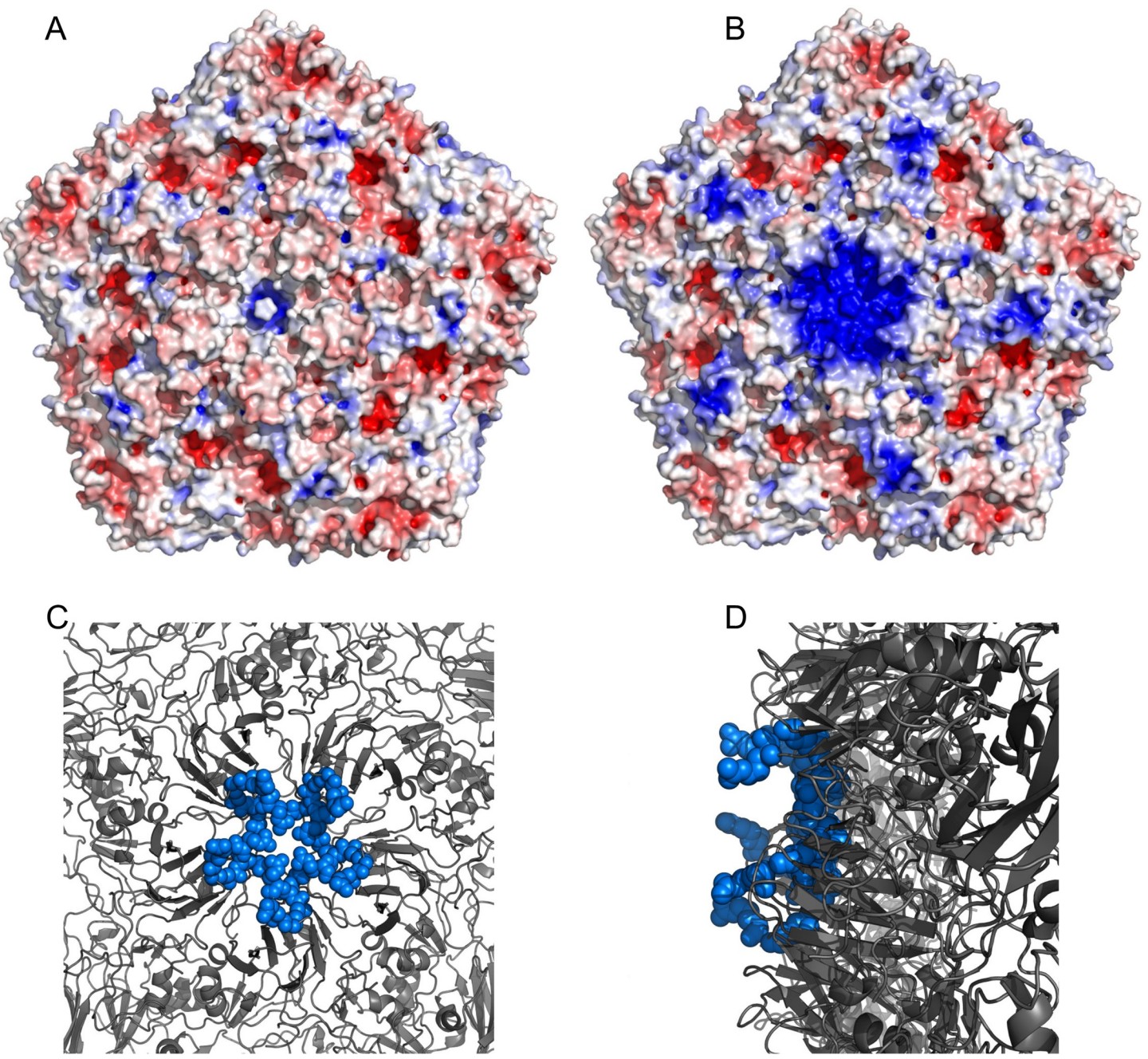

**Fig 3. Charge distribution and clustering of mutations on the capsid surface.** (**A**) Electrostatic potential of wild-type SAT1 capsid showing uniform charge distribution on the particle surface. (**B**) Electrostatic potential of the mutant SAT1 capsid showing clustering of positive charge at the 5-fold axis. The electrostatic potential was calculated using the APBS plugin embedded within Pymol. The colouring represents positive charge (blue), negative charge (red) and neutral (white). (**C**) The projection of the capsid on a 5-fold axis shows the clustering of the positively charged mutations from the top view. (**D**) The projection showing the side view and the surface exposure of the mutant side chains.

biological analysis. All recombinant progeny viruses maintained their chimeric integrity after seven passages in BHK-21 cells, as evidenced by RT-PCR and nucleotide sequencing. One-way antigenic relationships ($r_1$-values) were used to compare the antigenic cross-reactivity of the $v^{SAU}$SAT2 and its mutants to SAT2/SAU/6/00 vaccinated cattle antisera. The intra-serotype

$v^{SAU}$SAT2 progeny viruses were all neutralized by SAU/6/00 antisera ($r_1$-values 0.89–1.0) (S2 Table). Similarly, the inter-serotype viruses showed cross-reactivity to anti-SAT1 antisera similar to SAT1/NAM/307/98 ($r_1$-values 0.12–0.2). We also vaccinated cattle with a SAU/6/00$^{BHK58}$ vaccine and used the sera in VNTs. The sera neutralize both the original isolate (SAU/6/00$^{CT1}$) and SAU/6/00$^{BHK58}$ equally well ($r_1$-value = 1). The data indicated that the antigenic properties of serially passaged virus or the mutant viruses did not significantly change compared to the $v^{SAU}$SAT2 or $v^{NAM}$SAT2 viruses.

## Effect of the mutations on the infectivity and cell entry of cultured BHK-21 and CHO cells

To confirm the significance of the introduced repeat clusters of positive charge on the virion, the relative infectivity of the mutant viruses was determined in cultured cells (Table 3). The introduction of positively charged residues at amino acid residue positions 135 and 175 of SAT1 VP3 in the inter-serotype chimera viruses, rv$^{NAM}$VP3$^{135K}$ and rv$^{NAM}$VP3$^{135K,175K}$, resulted in the same medium (3–5 mm) to large (6–8 mm) clear plaque morphologies on BHK-21 cells as the wild-type $v^{NAM}$SAT2 virus, but no growth on CHO-K1 or any of the other CHO cell lines. Nevertheless, the rv$^{NAM}$VP3$^{135K}$ virus produced a 100-fold higher titre on BHK-21 cells, i.e. $2.51×10^8$ PFU/ml at passage level 7 compared to the SAT1/NAM/307/98 wild-type virus, i.e. $1.26×10^6$ PFU/ml at passage 8 on BHK-21 cells.

The $^{110}$KRR$^{112}$ of the rv$^{SAU}$VP1$^{110KRR}$ mutant was the only mutation with an appreciable effect on the growth of the intra-serotype chimeric virus. The mutant resulted in large clear plaques and a titre of 1000-fold higher ($3.98×10^8$ PFU/ml) than the wild-type virus in BHK-21 cells (Table 3). The intra-serotype chimera viruses with mutations in the VP1 N-terminus (rv$^{SAU}$VP1$^{50L,55N}$) or the βG-βH loop (rv$^{SAU}$VP1$^{158K}$) displayed medium (3–5 mm) to large (6–8 mm) opaque plaques on BHK-21 cells, and produced infectivity titres of $2.51×10^6$ PFU/ml and $6.31×10^6$ PFU/ml, respectively, which are similar to that of SAT2/SAU/6/00. However, small (1–2 mm) to medium clear plaques were observed on BHK-21 cells for the single mutants rv$^{SAU}$VP1$^{83K}$ and rv$^{SAU}$VP3$^{158K}$ with no discernible effect on virus growth in BHK-21

**Table 3. Titres (PFU/ml) of inter-serotype or intra-serotype FMDV chimera viruses with positive charged residue substitutions in the VP1 or VP3 capsid proteins.** Plaque morphology on BHK-21 cells are indicated.

| Virus mutants | Mutations in VP3 | Mutations in VP1 | Cell lines, plaque morphology and virus titres | | | | | |
|---|---|---|---|---|---|---|---|---|
| | | | BHK-21 | Plaque morphology | CHO-K1 | CHO-677[a] | CHO-745[b] | CHO-Lec2[c] |
| $v^{NAM}$SAT2 | - | - | $1.26×10^6$ | L, M, clear | N | N | N | N |
| rv$^{NAM}$VP3$^{135K}$ | E135K | - | $2.51×10^8$ | L, M, clear | N | N | N | N |
| rv$^{NAM}$VP3$^{135K,175K}$ | E135K, E175K | - | $7.94×10^6$ | L, M, clear | N | N | N | N |
| $v^{SAU}$SAT2 | - | - | $1.6×10^5$ | L, M, opaque | N | N | N | N |
| rv$^{SAU}$VP3$^{158K}$ | E158K | - | $1.26×10^5$ | M, S, opaque | $1.8×10^3$ | N | N | $1.4×10^3$ |
| rv$^{SAU}$VP1$^{50L,55N}$ | - | V50L-D55N | $2.51×10^6$ | L, M, opaque | N | N | N | N |
| rv$^{SAU}$VP1$^{158K}$ | - | T158K | $6.31×10^6$ | L, M, clear | $2.9×10^4$ | N | N | $1.4×10^4$ |
| rv$^{SAU}$VP1$^{83K}$ | - | E83K | $2.51×10^6$ | M, S, clear | $4.0×10^6$ | N | N | $1.6×10^6$ |
| rv$^{SAU}$VP1$^{83K,85R}$ | - | E83K, T85R | $7.94×10^4$ | M, S, clear | $1.4×10^4$ | N | N | $1.2×10^3$ |
| rv$^{SAU}$VP1$^{110KRR}$ | - | $^{110}$KGG$^{112}$-KRR | $3.98×10^8$ | L, clear | $4.7×10^8$ | N | N | $5.5×10^5$ |

[a] CHO cell mutant, heparan sulphate deficient (HS⁻) cell line.

[b] CHO cell mutant, heparan sulphate deficient (HS⁻) and chondroitin sulphate deficient (CS⁻) cell line.

[c] CHO cell mutant, sialic acid deficient (SA⁻) cell line.

N indicates no growth.

L is for large plaques (6–8 mm in diameter), M for medium plaques (3–5 mm in diameter) and S for small plaques (1–2 mm in diameter).

cells. Unexpectedly, the simultaneous replacement of E83 and T85 in the SAT2/SAU/6/00 VP1 protein with positive charged residues (mutant rv$^{SAU}$VP1$^{83K,85R}$) greatly reduced the replication capacity of the mutant virus and produced a titre of $7.94 \times 10^4$ PFU/ml. Five mutant viruses, *i.e.* rv$^{SAU}$VP3$^{158K}$, rv$^{SAU}$VP1$^{110KRR}$, rv$^{SAU}$VP1$^{83K}$, rv$^{SAU}$VP1$^{83K,85R}$ and rv$^{SAU}$VP1$^{158K}$, expanded their cell tropism and attained the ability to infect and replicate in CHO-K1 cells and the sialic acid-deficient (SA$^-$) CHO-Lec2 cells. However, none of the mutant viruses were able to infect the HSPG-deficient (HS$^-$) CHO-677 cells or the HS- and chondroitin sulphate (CS$^-$)-deficient CHO-745 cell line (Table 3).

To support the dependence of the mutant viruses on HSPG for cell entry, we used heparin as an inhibitor of virus growth. Mutant viruses were incubated with various concentrations of heparin prior to performing plaque assays on CHO-K1 cells. Plaque formation was completely abolished for the rv$^{SAU}$VP3$^{158K}$, rv$^{SAU}$VP1$^{83K}$ and rv$^{SAU}$VP1$^{158K}$ viruses at heparin concentrations of 0.625 mg/ml or higher (Fig 4A). Plaque formation of rv$^{SAU}$VP1$^{83K,85R}$ and rv$^{SAU}$VP1$^{110KRR}$ was linearly reduced with increasing concentrations of heparin until plaque formation was abolished for rv$^{SAU}$VP1$^{83K,85R}$ at 5 mg/ml heparin and, at 10 mg/ml, rv$^{SAU}$VP1$^{110KRR}$ had a titre of $0.7 \times 10^2$ PFU/ml (Fig 4A). This result suggests rv$^{SAU}$VP1$^{110KRR}$ may enter and replicate in CHO-K1 cells in the presence of high concentration of heparin.

To confirm the involvement of HSPG in virus binding, BHK-21 cells were treated with heparinase I or heparinase III prior to infection with intra-serotype chimeric viruses. The rv$^{SAU}$VP1$^{83K,85R}$ infection of BHK-21 cells without any heparinase treatment had a log titre of 6.3 but infection was completely blocked by heparinase treatment of the cells (Fig 4B). However, the infectivity of rv$^{SAU}$VP1$^{110KRR}$ was reduced by 22% (log titre of 5.6) in heparinase I-treated BHK-21 cells and by 18% (log titre 5.9) in heparinase III-treated BHK-21 cells when compared to the virus titres obtained in untreated BHK-21 cells (log titre 7.2) (Fig 4B).

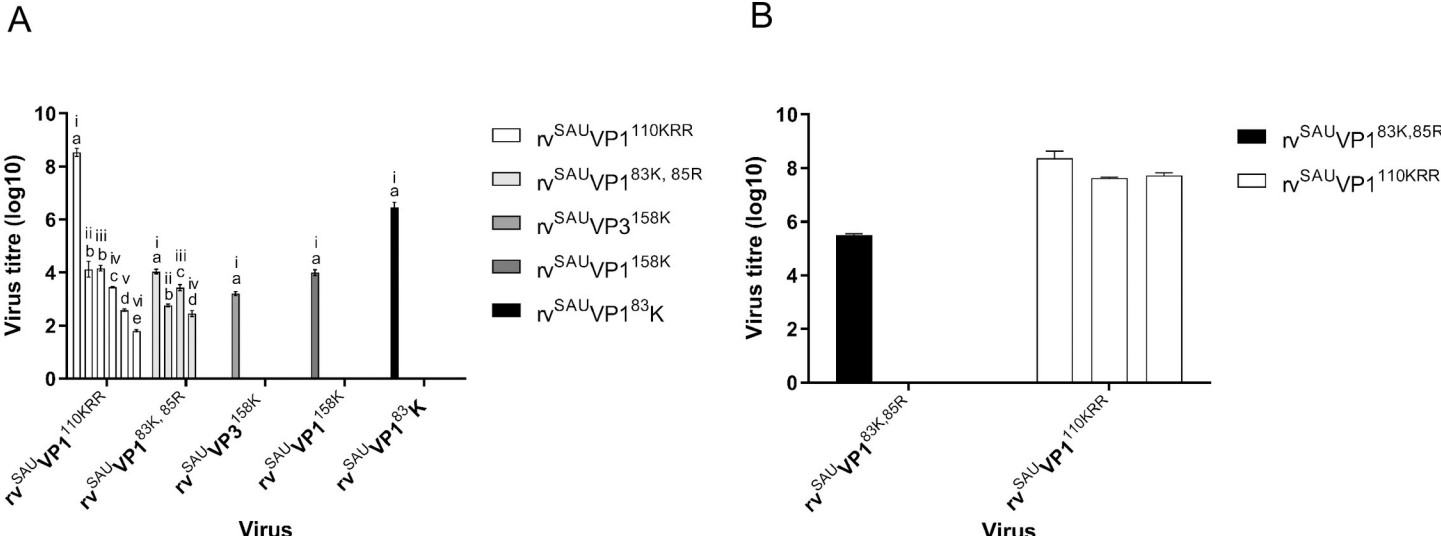

**Fig 4. Infection of CHO-K1 cells by the recombinant mutants rv$^{SAU}$VP3$^{158K}$, rv$^{SAU}$VP1$^{83K}$ and rv$^{SAU}$VP1$^{158K}$ is inhibited by heparin.** (A) Viruses with titres ranging from $5 \times 10^7$ to $7 \times 10^7$ PFU/ml were mock-treated or treated with different concentrations of soluble heparin, where i, ii, iii, iv, v and vi refers to 0mg/ml, 0.625mg/ml, 1.25mg/ml, 2.5mg/ml, 5mg/ml and 10mg/ml heparin respectively, before infecting CHO-K1 cell monolayers. Each treatment was performed in triplicate. Virus that had not been internalized was removed by washing with MES buffer (pH 5.5). The number of plaques formed on CHO-K1 cells was counted and expressed as the percentage of infectivity in relation to the non-heparin treated chimeric viruses. Statistical analyses were performed using a one-way ANOVA (followed by a Bonferroni's Multiple Comparison test). The confidence interval was 95%. Statistical analyses were carried out using GraphPad Prism v5.0 (GraphPad Software). Error bars represent the standard deviation. The a, b, c, d, or e denotations refer to statistically significant groupings where a is statistically significant to b, c, d and e; b is statistically significant to c, d and e; c is statistically significant to d and e and d is statistically significant to e. (B) Heparinase treatment of BHK-21 cells. BHK-21 cell monolayers were incubated with heparinase I or III enzymes or were mock-treated for 30 min at 37°C before virus infection (100 PFU/well). The number of plaques were determined at 24 hpi on BHK-21 cells and the percentage reduction in plaques calculated. The data represent means ± SD from three independent experiments.

Collectively, our results indicate that positively charged residues on surface-exposed loops of VP1 and VP3 are involved in the expansion of cell tropism of SAT viruses. However, not all the mutations resulted in a growth advantage in BHK-21 cells. Only positively charged residues in VP1 at positions 110–112 caused a 10- to 100-fold higher titre in BHK-21 cells for SAT1 viruses (our previous findings) and SAT2 viruses (Table 3).

## Discussion

Our findings suggest that binding to cell surface HSPG plays little or no role in the interaction between FMDV field isolates and host cells *in situ*. Nonetheless, SAT viruses rapidly mutate to expand their cell tropism and utilize HSPG during cell culture adaptation. Although infection with low-passage SAT1 and SAT2 viruses was predominantly HSPG-independent, several cell culture-adapted strains, each with increased clusters of positively charged residues at the 5-fold axis, attached to cells through an HSPG-dependent mechanism. We identified critical amino acids at VP1 residues 111–112 and 84–85 involved in defining this phenotype. Interestingly, none of the SAT1 and SAT2 viruses acquired a positive charge residue at position 56 of the VP3 protein, reported previously for serotypes O and A during cell culture adaptation [26,29]. This is likely to be due to divergence of the SAT viruses having eliminated the cryptic GAG site in those viruses.

Cytolytic passaging of FMDV in cell culture leads to rapid change in the preference of cell surface molecules for cell entry, which is made possible by the selective pressure exerted by the cell surface molecules [20,26,27,34]. To understand and exploit the dynamics involved in adaptation of SAT serotype viruses to cell culture, we showed that small, clear-plaque variants of SAT1 and SAT2 viruses are readily selected upon cytolytic passage in cell culture. Maree *et al*, [38] showed that plaques for SAT1 and SAT2 wild-type viruses are generally large with opaque edges and cell culture adaptation is accompanied by smaller to medium plaques and clear edges. HSPG usage was demonstrated by the reduction in the number of plaques in the presence of heparin and the inability of the viruses to infect HSPG-deficient cells. The results of this study indicated that most mutational patterns that arose during cytolytic passage of SAT viruses increased the net positive charge around the 5-fold axis of the virion, creating localized patches with the ability to bind a moiety of roughly the same size and charge of a sulphated glycan. Indeed, SAT viruses adopting such a mutation pattern were able to infect CHO-K1 cells, whilst plaque formation in BHK-21 cells was inhibited in the presence of soluble heparin as well as heparinase enzymes.

An increased number of surface positive charges after cytolytic passage in cell culture has been correlated with the ability to infect CHO-K1 cells and the use of HS receptors, and has been observed for a number of diverse virus families. Coxsackievirus A9 [42], herpesviruses [43, 44], cytomegalovirus [45], HIV [46,47], pseudorabies virus [48], equine arteritis virus [49], dengue virus [50], porcine reproductive and respiratory syndrome virus [51], classical swine fever virus [52], respiratory syncytial virus [53], sindbis virus [54], tick-borne encephalitis [55], vaccinia virus [56] and adeno-associated virus [57] have all been reported to select mutants with high affinity for binding to GAGs, especially HS, after repeated passage in cultured cells. For FMDV serotypes O and A residues from both VP1 and VP3 contribute to exposed positively charged clusters on the viral capsid [28,29]. We found an analogous situation for SAT1 and SAT2 viruses with clusters of two to four basic amino acids (lysine or arginine). The capsid proteins of the SAT1 and SAT2 viruses analysed here differed by 34–36% and 32–34% of their amino acid positions from the serotype $A_{61}10$ and $O_1BFS$ viruses [28,29] in any pairwise alignment.

The question arises as to whether the structural requirements for HS binding sites on complex protein structures, like FMDV, are created by chance as a result of random mutations or

whether a structurally predisposed binding site exists on the capsid [28]. Analysis of the capsid region of non-HSPG binding strains suggests that a single basic residue, although rare, may occur in the VP1 βD-βE or βF-βG loops of SAT1 and SAT2 viruses; however, on its own it is insufficient to support HSPG interaction. A mutation of a spatially closely positioned residue that may be associated with optimization of cell entry will provide a selective advantage and will be strongly selected for [20,37,58–60]. VP1 residue positions 111–112 and 84–85 are both unique as they are arranged exposed on the surface close to the 5-fold axes. Our data suggest that clustering of positively charged residues at the 111–112 location enables interaction with HS. This implies that the virus particle has 12 HSPG binding patches, each located at a vertex of the icosahedron.

To confirm an HSPG-binding role for residues at position 111–112, we have recently reported on the construction of two infectious SAT1 genome-length clones, with substitutions of residues 111–112 to positively charged amino acids [36,38]. Both recombinant SAT1 viruses were capable of infecting CHO-K1 cells, suggesting infection via HSPG-binding. This is in agreement with the findings in a type A virus, where a single positively charged substitution at residue 110 of VP1 resulted in the ability to infect CHO-K1 cells [34].

Phenotypic characterisation of the recombinant mutants revealed that the SAT2 variants with positively charged residues in VP1 or VP3 surface-exposed loops were able to infect CHO-K1 cells and plaque formation was inhibited by the presence of heparin. The growth of the engineered SAT2 single mutants 158K and 83K in VP1 and 158K in VP3 were completely inhibited by soluble heparin. However, none of these mutants showed any growth advantage on BHK-21 cells, compared to the wild-type chimeric v$^{SAU}$SAT2 virus. Our results suggest that KRR residues at positions 110–112 of VP1 are the most important residues for cell culture adaptation and expansion of cell tropism for SAT1 [38] and SAT2 viruses (Table 3). These mutants had enhanced infectivity of BHK-21 cells. The combined K and R residues at positions 83 and 85 of VP1 also extended cell tropism of mutated FMDV and may even enhance infectivity of BHK-21 cells in certain circumstances, as observed for isolates that underwent repeated cytolytic passages on BHK-21 cells. Nevertheless, in mutational studies the rv$^{SAU}$VP1$^{83K,85R}$ showed poor performance in BHK-21 cells despite the extended cell tropism. This highlights the unique position of residues 110–112 of VP1 as no other amino acid position in the P1 region can generate such a concentration of positively charged residues on the capsid. The interaction between HSPG, its cellular ligands and viruses is mostly electrostatic and the ability to infect CHO-K1 cells is tolerant to either R or K at VP1 position 110–112 [34,38,43,61].

It remains possible that other residues contribute to HSPG binding in SAT1 and SAT2 viruses. Examples from our study include the VP1 W87R substitution of SAT1/KEN/5/98 that appeared simultaneously with the VP1 E84K mutation, and VP1 N48K together with the VP1 N111K substitution in SAT1/NIG/5/81. Both are spatially close to the VP1 βD-βE loop and contribute to the positively charged cluster around the 5-fold axis. In several cases, the disappearance of negatively charged amino acids or the appearance of residues with a partial positive charge has been observed in association with lysine or arginine residues. Although we found two examples of positively charged substitutions (VP2 Q74R and E133K) in the shallow depression at the junction of the three major capsid proteins, as described for type A and O viruses [28,29], the role of these substitutions in the interaction of SAT1 capsids to HSPG is still unclear.

The introduction of positively charged residues clustered on the 5-fold vertices of SAT1 and SAT2 viruses pose the question whether tissue culture propagation may adversely affect vaccine seed through the selection of viruses altered at immunological important sites. However, the antigenicity of v$^{SAU}$SAT2 and the mutant viruses seemed to be unchanged based on

limited cross neutralization data using SAT2/SAU/6/00 antisera. Likewise, a vaccine produced after SAT2/SAU/6/00 was passaged 58 times through BHK-21 cells protected cattle against challenge with the outbreak isolate (S2 Table).

This study further strengthens the knowledge of the molecular mechanisms during expanded cell tropism of SAT viruses. FMDV has been shown to utilize other non-integrin and non-HS receptors [21,25,62]. Serotype O and C viruses that lack the RGD motif have been shown to infect HS-deficient cells [20–22,24,25,61]. Berryman *et al.* [34] showed that a single amino acid substitution at VP1-110 (VP1 Q110K) allowed for HS-independent infection of CHO-677 cells. Additionally, Chamberlain *et al.* [24] provided evidence that residue changes in VP2 130–131 E to K account for the cell culture adaptation and extended cell tropism of a variant of FMDV. Our results show that two SAT viruses (SAT1/UGA1/97 and SAT2/UGA/2/02) were able to infect CHO-677 (HS⁻) and CHO-745 (HS⁻, CS⁻) cells, indicating cell entry independent of sulphated proteoglycans or $\alpha_V$-integrin attachment. SAT1/UGA1/97 and SAT2/UGA/2/02 infectivity in CHO-K1 cells was also unaffected by the presence of heparin (Table 1). SAT1/UGA1/97 had VP2 E133K and VP1 E58K positively charged substitutions occurring together with two other significant surface-exposed changes (L115Q in VP2 and H46N in VP1), whilst SAT2/UGA/2/02 revealed seven amino acid changes with only one positively charged substitution, *i.e.* VP1 E83K (S1 Table). Chamberlain *et al.* [24] reported that the changes associated with cell culture adaptation of a serotype A virus resulted in a net gain of positive charges in local regions of the viral capsid; however, infectivity of CHO-677 cells suggested a HS-independent mechanism for cell entry. We postulate that SAT1/UGA/1/97 and SAT2/UGA/2/02 may utilize similar HS-independent mechanisms to enter cells. Lawrence *et al.* [25] demonstrated that FMDV with a VP1 E95K/S96L and an RGD to a KGE substitution in the VP1 βG-βH loop are involved in the interaction with the cell membrane molecule, Jumonji C-domain containing protein 6 (JMJD6) during cell entry. In addition, FMDVs have been shown to utilize the macropinocytosis pathway to infect cells [25,58]. The proposed macropinocytic pathway involves the binding of FMDV to receptor tyrosine kinases (RTK). In summary, as indicated many years ago by Mason *et al.* [14], specific receptor-mediated conformational changes are not required for FMDV to infect cells, it is sufficient to internalize the particle and provide a modest drop in pH to trigger uncoating.

Adaptation of FMDV to non-host cells, like cultured hamster cells, is a common step to expand the virus cell tropism during the vaccine production process, but is time-consuming and often unsuccessful [24,63,64]. Knowledge of the HSPG-binding sites on FMDV and the role HSPG plays in assisting cell entry can be applied in the construction of chimeric viruses containing the symmetrical, positively charged clusters that enable interaction with HSPG. We found that a cluster of positively charged residues at positions 110–112 of VP1, are symmetrically arranged around the 5-fold axis of the virion and are the most important residues for cell culture adaptation and expansion of cell tropism for SAT1 and SAT2 viruses. This might aid in avoiding the accumulation of random changes that lead to divergence of the vaccine strain from the virus populations circulating in nature. This study further emphasizes that much remains unknown regarding FMDV adaptation and cellular receptors for cell entry.

## Materials and methods

### Cells, viruses and plasmids

Baby hamster kidney (BHK) cells, strain 21, clone 13 (ATCC CCL-10), used during virus passage and plaque assays, was maintained in Glasgow minimum essential medium (GMEM, Sigma-Aldrich), supplemented with 10% (v/v) foetal bovine serum (FBS, Hyclone), 1× antibiotic-antimycotic solution (Invitrogen), 1 mM L-glutamine (Invitrogen) and 10% (v/v) tryptose

phosphate broth (TPB, Sigma-Aldrich). Wild-type Chinese hamster ovary (CHO) cells, strain K1 (ATCC CCL-61), were grown in Ham's F-12 nutrient medium (GIBCO), supplemented with 10% (v/v) FBS and 1% (v/v) antibiotics. The same maintenance medium was used for the CHO-K1 derivative cell lines, CHO-677 (pgsD-677) (ATCC CRL-2244), which is HSPG deficient (HS⁻), and CHO-745 (pgsA-745) (ATCC CRL-2242), which is HS⁻ and chondroitin sulphate (CS⁻) deficient. The CHO-Lec2 (Pro-5WgaRII6A) (ATCC CRL-1736) cell line, which is sialic acid (SA⁻) deficient, was grown in Alpha minimum essential medium (MEMAlpha, GIBCO) supplemented with 10% (v/v) FBS and 1% (v/v) antibiotics.

Fifteen FMDV isolates, belonging to the SAT1 and SAT2 serotypes, were included in this study. These viruses were either supplied by the World Reference Laboratory (WRL) for FMD at the Institute for Animal Health, Pirbright (United Kingdom), or are part of the virus bank at the Agricultural Research Council (ARC), Onderstepoort Veterinary Institute (OVI) (South Africa). The viruses were collected from either cattle or wildlife (Impala, *Aepyceros melampus* and African buffalo, *Syncerus caffer*), and were isolated on either primary pig kidney or bovine thyroid cells, followed by amplification on Instituto Biologico Renal Suino-2 (IB-RS-2) cell monolayers. Of the 15 virus isolates, 14 viruses were serially passaged eight times on BHK-21 cells and the SAT2/SAU/6/00 virus (calf thyroid passage one, SAT2/SAU/6/00$^{CT1}$) was serially passaged 58 times on BHK-21 cells.

The nucleotide sequences generated for the 15 cell culture-adapted FMDV isolates were submitted to GenBank. The respective accession numbers are as follows: GU194495 (SAT1/KNP/148/91), GU194498 (SAT1/KNP/41/95), GU194497 (SAT1/ZIM/13/90), DQ009721 (SAT1/KEN/05/98), AF378302 (SAT1/TAN/1/99), AY442012 (SAT1/UGA/01/97), DQ009725 (SAT1/SUD/03/76), DQ009723 (SAT1/NIG/5/81), DQ009724 (SAT1/NIG/15/75), GU194502 (SAT1/NIG/06/76), GU194488 (SAT2/KNP/02/89), GU194489 (SAT2/KNP/51/93), AF367113 (SAT2/ZIM/10/91), DQ009731 (SAT2/UGA/02/02) and AY297948 (SAT2/SAU/6/00).

The construction of infectious genome-length plasmids pSAT2, p$^{NAM}$SAT2 and p$^{SAU}$SAT2 (superscript indicates the donor capsid-coding sequence) has been described previously [36,38,60,65]. In order to construct the chimeric cDNA clones, the outer capsid-coding region of pSAT2 was replaced with the corresponding regions of SAT1/NAM/307/98 or SAT2/SAU/6/00 by making use of the flanking unique restriction enzyme sites *Ssp*I and *Xma*I in the VP2 and 2A-coding regions, respectively [37,65]. The viruses recovered from pSAT2, p$^{NAM}$SAT2 and p$^{SAU}$SAT2 recombinant plasmid DNA were designated vSAT2, v$^{NAM}$SAT2 (inter-serotype chimera) and v$^{SAU}$SAT2 (intra-serotype chimera).

## Plaque titration

Titrations were performed by making use of a standard plaque assay method. Monolayers of BHK-21, CHO-K1, CHO-677, CHO-745 or CHO-Lec2 cells in 35-mm cell culture plates (Nunc) were infected with serially diluted viruses for 1 h, followed by the addition of a 2 ml tragacanth overlay [66,67] and incubation for 48 h or 72 h at 37°C. The overlaid, infected monolayers were stained with 1% (w/v) methylene blue in 10% ethanol and 10% formaldehyde in phosphate-buffered saline (PBS) (pH 7.4). Virus titres were calculated and expressed as plaque forming units per millilitre (PFU/ml).

## Virus neutralization test

Two-dimensional virus neutralisation tests were also carried out using SAU/6/00$^{BHK58}$ twice vaccinated cattle sera and following established methodology [23,36,38]. Antibody titres were calculated from regression data as the log10 reciprocal antibody dilution required for 50%

neutralisation of 100 tissue culture infective units of the test virus ($\log_{10}SN_{50}/100\ TCID_{50}$). The $r_1$-values were calculated as the ratio between the heterologous (mutants) and homologous (SAU/6/00) serum titres.

## RNA extraction, cDNA synthesis, PCR amplification and nucleotide sequencing

RNA was extracted from 200 μl infected cell lysates using a guanidium-based nucleic acid extraction method [68] and utilized as templates for cDNA synthesis. Viral cDNA was synthesized with SuperScript III (Life Technologies) and oligonucleotide 2B208R [69]. The *ca*. 3.0 kb leader and capsid-coding regions of the viral isolates were obtained by PCR amplification using Expand Long template *Taq* DNA polymerase (Roche) and SAT genome-specific oligonucleotides [36]. The consensus nucleotide sequences of the amplicons were determined using a primer-walking approach and the ABI PRISM BigDye Terminator Cycle Sequencing Ready Reaction Kit v3.0 (Perkin Elmer Applied Biosystems). The extension products were resolved on an ABI 3100 Genetic Analyzer (Applied Biosystems). Sequences were compiled and edited using Sequencher v5.4.6 (Gene Codes Corporation, Ann Arbor, MI, USA) sequence analysis software for Windows. The nucleotide and deduced amino acid sequences were aligned with ClustalX [70].

## Site-directed mutagenesis and sub-cloning

The P1-2A region of SAT1/NAM/307/98 and SAT2/SAU/6/00 was cloned into pBlueScript (Stratagene) and the respective recombinant plasmids were designated pNAM-P1 and pSAU-P1. Mutagenesis primers that were complementary to the P1 regions, are summarized in Table 4.

The QuikChange II XL Site-Directed Mutagenesis Kit (Stratagene) enabled the introduction of the desired mutations by an inverse PCR method, according to the manufacturer's instructions. The resulting plasmid DNA amplicons were transformed into XL10-Gold Ultra competent cells (Stratagene). The extracted plasmids were characterized by restriction enzyme digestion, followed by automated sequencing and selection of plasmid DNA containing the respective mutations.

**Table 4. Mutagenesis and In-Fusion cloning primer sequences.**

| Virus | Protein-coding region | Codon | Primer sequences |
|---|---|---|---|
| NAM/307/98 | | | |
| | VP3 | 135 | 5'-GGCACAAACCCTCTCCCC**a**AAACACCGGAGATGGCATC |
| | VP3 | 175 | 5'-CCTACACCTACGCTGAC**a**AGCCTGAACAGGCTTCAG |
| SAU/6/00 | | | |
| | VP3 | 158 | 5'-GCCGCGCACTGCTATCACGCG**a**AATGGGACACTGGACTGAACTC |
| | VP1 | 158 | 5'-GTACGCTGACAGCA**a**GCAC**a**CTTTGCCGTCAACCTTC |
| | VP1 | 50/55 | 5'-AACATCCTTT**c**TTGTGGACCTCATG**a**ACACAAAGGAGAAG |
| | VP1 | 83 | 5'-GTGGGC**a**A**a**CACCGGCGCGCCTTTTGGCAGCCTAAC |
| | VP1 | 83/85 | 5'-CTTGAGATTGCATGTGTGGGC**a**A**a**CAC**cggc**G**c**GCCTTTTGGCAGCCTAAC |
| | VP1 | 110–112 | 5'-GACAACCCCATGGTTTTCGCC**AAacG**ac**GTGTGACCCGCTTTGCCATCC |
| SAT2 In-Fusion[a] | | | |
| | VP2 | | 5'-*GCTCGAGGACCGAATATT*GACCACACGTCACGGAACCACGA |
| | VP1/2A | | 5'-*AGAAGAAGGGCCCGGGG*TTGGACTCAACGTCTCCTGCCT |

[a] The underlined sequence is complementary to the ends of the digested pSAT2.

The mutated P1-2A region of pNAM-P1 was cloned into the corresponding region of pSAT2 using the unique restriction sites *Ssp*I and *Xma*I. Cloning and transformation procedures were performed as described previously [36, 38]. For the mutated P1-2A region of pSAU-P1, cloning into the corresponding region of pSAT2 was achieved with the In-Fusion HD Cloning Kit (Clontech). Briefly, pSAT2 was digested with *Ssp*I and *Xma*I. A PCR primer set was designed for the mutated P1-2A region of pSAU-P1 with 15-bp extensions that were complementary to the ends of the digested pSAT2 (Table 4). Following PCR amplification with the Advantage 2 PCR Kit (Clontech), the resulting amplicons were purified with the Zymoclean GEL DNA recovery Kit (Zymo Research) from an agarose gel. The In-Fusion cloning reaction, containing the purified amplicon and the digested vector, was transformed into Stellar competent cells (Clontech). Clones containing the desired mutations were selected and designated $p^{NAM}VP3^{135K}$, $p^{NAM}VP3^{135K,175K}$ (inter-serotype), $p^{SAU}VP3^{158K}$, $p^{SAU}VP1^{50L,55N}$, $p^{SAU}VP1^{83K}$, $p^{SAU}VP1^{83K,85R}$, $p^{SAU}VP1^{110KRR}$ and $p^{SAU}VP1^{158K}$ (intra-serotype).

### *In vitro* RNA synthesis, transfection and virus recovery

The $p^{NAM}SAT2$, $p^{NAM}VP3^{135K}$, $p^{NAM}VP3^{135K,175K}$, $p^{SAU}SAT2$, $p^{SAU}VP3^{158K}$, $p^{SAU}VP1^{50L,55N}$, $p^{SAU}VP1^{83K}$, $p^{SAU}VP1^{83K,85R}$, $p^{SAU}VP1^{110KRR}$ and $p^{SAU}VP1^{158K}$ plasmids were linearized by digestion with *Swa*I and utilized as templates for *in vitro* RNA synthesis with the MEGAscript T7 Kit (Ambion). The RNA integrity was analyzed with agarose gel electrophoresis and then quantified spectrophotometrically. *In vitro* synthesized RNA (*ca*. 3 μg) was mixed with Lipofectamine2000 (Invitrogen) and incubated for 20 min at room temperature, prior to being transfected into BHK-21 cell monolayers prepared in 35-mm cell culture plates. Incubation was continued in GMEM containing 1% (v/v) FBS and 25 mM HEPES for 48 h at 37˚C with a 5% $CO_2$ influx.

Virus-containing supernatants that were freeze-thawed and clarified by centrifugation to remove cell debris was added (250μl) to fresh BHK-21 monolayers in 35-mm cell culture wells and incubated for 48 h at 37˚C. This cycle of passage continued until the cytopathic effect (CPE) observed was between 80 and 100%. Once viable viruses were recovered, automated sequencing was conducted to verify the presence of the engineered mutations.

### Heparin plaque reduction assay

Viruses with titres ranging from $5 \times 10^7$ to $7 \times 10^7$ PFU/ml were prepared in $1 \times$ PBS (pH 7.4) and added in a ratio of 1:1 to $1 \times$ PBS containing heparin (Sigma-Aldrich). Viruses were incubated in the presence of heparin for 30 min at room temperature. Heparin concentrations of 0.625 mg/ml, 1.25 mg/ml, 2.5 mg/ml, 5 mg/ml and 10 mg/ml were used, and each treatment was performed in triplicate. Following incubation, 500 μl of the virus-heparin mixture was added to sub-confluent CHO-K1 cell monolayers and virus was allowed to attach to the cells for 15 min at room temperature. The cell monolayers were then washed with $1 \times$ PBS (pH 7.4) and incubated at 37˚C for a further 15 min in Ham's F-12 nutrient medium containing 1% (v/v) FBS and 25 mM HEPES to enable virus internalization. Virus that had not been internalized was removed by washing with acidic MES (25 mM *N*-morpholino ethanesulfonic acid, pH 5.5, in 145 mM NaCl). Following a final wash step, 2 ml of tragacanth (Sigma) overlay was added and the monolayers were incubated at 37˚C for 48 h. Subsequently, the cells were fixed with formaldehyde and stained with 1% methylene blue. Plaques were counted and virus titers (PFU/ml) were determined. Statistical analyses were performed using a one-way ANOVA (followed by a Bonferroni's Multiple Comparison test). The confidence interval was 95%. Statistical analyses were carried out using GraphPad Prism v5.0 (GraphPad Software).

## Heparinase assay

Monolayers of BHK-21 cells in 24-well tissue culture plates were washed three times with GMEM and treated with either heparinase I or III (Sigma-Aldrich) in $1 \times$ PBS for 30 min at 37˚C. After enzyme treatment, the cells were washed three times with GMEM and $rv^{SAU}VP1^{83,85}$ or $rv^{SAU}VP1^{110KRR}$ viruses (100 PFU) were adsorbed onto the cells for 1 h at 37˚C. Following a final wash, complete GMEM was added and the cells were incubated at 37˚C for 24 h. Virus titres were determined by plaque assays on BHK-21 cells. The data was analysed from three independent experiments.

## Structural modelling and ligand docking

The three-dimensional structures of a FMDV SAT1 (2WZR) and a SAT2 capsid (5ACA) was utilized as a template for the prediction of a protein homology model of the SAT2 virus. This was built manually using the crystallographic object-oriented toolkit (COOT) [71] to correct the sequence and avoid clashes between symmetry related protomers, based on a sequence alignment generated using CLUSTALW [72]. Only the local geometry was refined in COOT. Pentameric models were generated for each of the SAT1 and SAT2 structures using the non-crystallographic symmetry. In addition, the cell culture-adapted mutations were modelled using COOT to make mutant virus structures for both SAT1 and SAT2. Structures were visualized with RIVEM [40] and PyMol v0.98 (DeLano Scientific LLC), and the electrostatic surface potential was calculated using the APBS module of PyMol. GRID [41] was used to identify an energetically favourable binding site of a heparan sulphate moiety. The GRID calculation was performed within a 20 Å radius of the 5-fold axis using a probe appropriate to sulphate. Five linked disaccharide molecules corresponding to the observed heparin, bound to FMDV O1BFS (1QQP) [29], were docked onto the SAT1 pentamer (both the wild-type and cell-adapted model) using GOLD 5.1 (Cambridge Crystallographic Data Centre) with default parameters. The best docking 'poses' were judged by manual inspection and assessing the GOLD fitness score.

## Supporting information

**S1 Table. Summary of the amino acid substitutions in the outer capsid proteins of SAT1 and SAT2 viruses resulting from cytolytic passages in BHK-21 cells.**
(DOC)

**S2 Table. Virus neutralization and predicted cross-reactivity of SAT2/SAU/6/00 vaccinated cattle sera to the isolates and chimeric mutant viruses.**
(DOC)

## Acknowledgments

The SAT1 and SAT2 viruses were either supplied by the World Reference Laboratory (WRL) for FMD at the Institute for Animal Health, Pirbright (United Kingdom), or received from the regional FMD reference laboratory at Transboundary Animal Diseases of the ARC-OVI. We are indebted to Drs B. Charleston, E. Rieder and T. Jackson for fruitful collaboration and discussions. We would also like to thank Drs B. Mans and O. Koekemoer for critical reading of the manuscript. Many thanks to Dr K.A Scott for assistance with the one-way ANOVA analysis.

## Author Contributions

**Conceptualization:** Melanie Chitray, Guntram Paul, Jacques Theron, Elizabeth E. Fry, David I. Stuart, Francois F. Maree.

**Data curation:** Melanie Chitray, Abhay Kotecha, Peninah Nsamba, Jingshan Ren, Sonja Maree, Tovhowani Ramulongo, Jacques Theron, Francois F. Maree.

**Formal analysis:** Melanie Chitray, Abhay Kotecha, Peninah Nsamba, Sonja Maree, Elizabeth E. Fry, David I. Stuart, Francois F. Maree.

**Funding acquisition:** Guntram Paul, Elizabeth E. Fry, David I. Stuart, Francois F. Maree.

**Investigation:** Melanie Chitray, Abhay Kotecha, Peninah Nsamba, Jingshan Ren, Sonja Maree, Tovhowani Ramulongo, Elizabeth E. Fry, David I. Stuart, Francois F. Maree.

**Methodology:** Melanie Chitray, Abhay Kotecha, Peninah Nsamba, Jingshan Ren, Sonja Maree, Tovhowani Ramulongo, Jacques Theron, Elizabeth E. Fry, David I. Stuart, Francois F. Maree.

**Project administration:** Melanie Chitray, Abhay Kotecha, Elizabeth E. Fry, David I. Stuart, Francois F. Maree.

**Resources:** Melanie Chitray, Guntram Paul, Jacques Theron, Elizabeth E. Fry, David I. Stuart, Francois F. Maree.

**Software:** David I. Stuart.

**Supervision:** Jacques Theron, Elizabeth E. Fry, David I. Stuart, Francois F. Maree.

**Visualization:** Jacques Theron, Elizabeth E. Fry, David I. Stuart, Francois F. Maree.

**Writing – original draft:** Melanie Chitray, Abhay Kotecha, Peninah Nsamba, Jingshan Ren, Sonja Maree, Tovhowani Ramulongo, Jacques Theron, Elizabeth E. Fry, David I. Stuart, Francois F. Maree.

**Writing – review & editing:** Melanie Chitray, Abhay Kotecha, Guntram Paul, Jacques Theron, Elizabeth E. Fry, David I. Stuart, Francois F. Maree.

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
