## [Decision Letter · Decision Letter 0]

15 Mar 2020

Dear Dr. Maree,

Thank you very much for submitting your manuscript "Symmetrical arrangement of positively charged residues around the 5-fold axes of SAT type foot-and-mouth disease virus enhances cell culture of field viruses" for consideration at PLOS Pathogens. As with all papers reviewed by the journal, your manuscript was reviewed by members of the editorial board and by several independent reviewers. In light of the reviews (below this email), we would like to invite the resubmission of a significantly-revised version that takes into account the reviewers' comments.

Each of the reviewers makes specific comments that should be addressed in a revised version of the manuscript. In particular, we would like you to address the novelty of this work and its suitability for PLoS Pathogens.

We cannot make any decision about publication until we have seen the revised manuscript and your response to the reviewers' comments. Your revised manuscript is also likely to be sent to reviewers for further evaluation.

Sincerely,

Richard J. Kuhn, PhD

Associate Editor

PLOS Pathogens

Michael Diamond

Section Editor

PLOS Pathogens

Kasturi Haldar

Editor-in-Chief

PLOS Pathogens

orcid.org/0000-0001-5065-158X

Michael Malim

Editor-in-Chief

PLOS Pathogens

orcid.org/0000-0002-7699-2064

Each of the reviewers makes specific comments that should be addressed in the revised version of the manuscript. In particular I would like you to address the novelty of this work and its suitability for PLoS Pathogens.

Reviewer's Responses to Questions

**Part I - Summary**

Reviewer #1: FMDVs remain as significant threats to poultry and livestock industry. Developing vaccines from emerging new strains are difficult due to that some of the field viruses are unable to be adapted to cell cultures. This study analyzed the mutations of cell-culture-adapted field viruses. Many of these mutations are positively charged residues clustered around the icosahedral five-fold axis. The authors then designed a series of experiments to show that these mutated viruses have adapted new host receptors, heparin sulfate proteoglycan (HSPG). They also generated chimera viruses that mimic those observed mutations. These chimera viruses were able to adapt to cell culture well, which provide a novel way to quickly adapt emerging field viruses to cell culture for vaccine developments.

The molecular virology experiments are thoroughly designed, and the results are rich and convincing. Many different aspects have been tested and discussed. This part is the strength of the paper.

The molecular simulation docking is OK but not as convincing as the other experiments. With highly positively charged 5-fold, the negatively charged heparin will be docked there. The most convincing results will be a structure of the virus/heparin complex. With current cryo-EM techniques, obtain a complex structure using purified virus and heparin can be done very quickly. No need to obtain atomic resolution. Extra densities on the five-fold positively charged areas will prove the virus has adapted heparin as the receptor if using purified virus incubate with heparin. The author team has an expert in using cryo-EM to solve picornavirus structures.

In summary, the paper is a very good one with a lot of convincing data from lab experiments. The simulation part is not necessary. Completely remove the simulation will not affect the significance of the discovery. Normally simulation can provide useful hints or suggestions for designing lab experiments. Here, the reviewer does not see that. Simulation itself cannot be a strong data to prove the receptor binding.

Reviewer #2: Symmetrical arrangement of positively charged residues around the 5-fold axes of SAT type foot-and-mouth disease virus enhances cell culture of field viruses.

FMD viruses of Southern African Territories (SAT) serotypes were adapted to growth on the BHK cell line. Adapted viruses gained the ability to enter cells by interaction with heparan sulphate proteoglycan (HSPG) via the selection of synonymous substitutions resulting in increased positive charge around the 5-fold symmetry axis of the viral capsid. The most important residues for adaptation were confirmed by characterising modified viruses generated by reverse genetics.

This is a meticulous study by well-respected authors which has generated robust data which adds some new information to the process of culture adaptation of FMDV.

Reviewer #3: The submitting groups here have a long background in FMDV and the manuscript is correspondingly authoritative. In the main the data hangs together well. The subject is the more problematic members of the FMDV serotypes, SAT1 and SAT2, generally regarded as difficult to grow and/or unstable. Selection of rapidly growing varants from the quasispecies by extensive passage results in isolates that bypass the integrin receptors and enter cells by the more generic mechanism of HS binding, as has been published for other serotypes. These viruses are characterized and reveal a predominance of charge changes with a subset of these clustered around the 5 fold axis. It is assumed the avidity of the cluster accounts for the new binding properties and to an extent this is confirmed by substitutions in molecular clones which indicate one (at 111-112 in VP1), sometimes with other surface exposed changes, are the key charged required. The knock-on of this is that field isolates required to be grown for vaccine use could be “fixed” by the introduction of the key change before being scaled up. This is useful but I did get the sense the story had been decided before the data were put together and a few more caveats might be wise. For example:

I lost the connection, if there is one, with the mutation and plaque morphology. While size and opacity relate to virus yield they are also affected by cpe and cell to cell transmission. There is still quite a range of titres in the adapted viruses. Are we sure it's all just better entry=better yield?

I think the exceptions illustrated by 158K and 83K in VP1 and 158K in VP3 need a little more weight. They are inhibited by heparin but do not go in BHKs. There is a trend in the manuscript to the view that +ve charge=HS binding=enhanced entry=enhanced yield. But these mutants do not fit - why not? What’s not right about them?

I think the application should be more reserved. Only two of the final mutants reach titres (>10^8) that would be respectable enough for mass culture and it remains to be seen how many field isolates would do the same when engineered in this way.

It seems probable that the mutants here would generate the same antibody response on immunisation at the WT but it is not yet proven. It's possible that they could affect antigen processing.

There are recombinant routes to FMDV vaccines, the rec.Adeno for example, where the need to culture is bypassed anyway. Would these mutations benefit or hinder such approaches?

**Part II – Major Issues: Key Experiments Required for Acceptance**

Reviewer #1: No major issues have been found. However, the reviewer suggests the authors to add discussions about whether making those chimeric viruses with positive charged residues clustered on the 5-fold vertices might alter the antigen properties of the viruses. The animal might develop immune response to those areas which might defeat the purposes of vaccine development of emerging field strains.

Reviewer #2: This is a meticulous study by well-respected authors which has generated robust data which adds some new information to the process of culture adaptation of FMDV. However, the main claims of the study are already very well established. References (cited in the manuscript) 21, 44, 45 and 23, all describe previous culture adaptation of FMDV involving acquisition of positive charge around the five-fold axis. Therefore, the novelty of the current study is not what would normally be consistent with publication in PLoS Pathogens.

What is of more concern is that the introduction and discussion does not make this situation clear. Despite the references being cited within the paper, there is no explanation in the introduction, of these previous studies (e.g. references 21, 44, 45, 23) having already identified acquisition of positive charge around the 5-fold as a mechanism of culture adaptation. This seems bizarre. Instead the following statement is included:

“The genetic alterations associated with increased cell killing activity during cytolytic passages of the SAT serotype FMD viruses in BHK-21 cells are largely unknown. Nevertheless, it has been noted that amino acid substitutions accumulate in their capsids during serial passaging (44–46)”

One of the references at the end of this statement is even from one of the authors of the current study and specifically describes the acquisition of positive charge around the five-fold as a mechanism of culture adaptation of SAT type viruses. In the context of the novel claims of the current manuscript this statement and other omissions appear misleading. Perhaps hopefully all of this is the result of lack of oversight, but I’m afraid it is rather disappointing.

Reviewer #3: None

**Part III – Minor Issues: Editorial and Data Presentation Modifications**

Reviewer #1: 1. The paper still contains some typos that need to be read carefully and correct. Eg. In Ln177 “the roadmap programme”, the Roadmap should be capitalized to be consistent with Ln534. Another example in Ln263 “heparinise” should be “heparinase”?

2. The authors need to check all the references carefully. Some of the references are not correctly cited. For example, for the same Ln177 mentioned above. Reference 48 is not the Roadmap program but the program GRID for docking as shown in Ln191. Again, on Ln534, the Roadmap was cited as reference 47, which is a very old review paper for virus-polysaccharide interactions. None of the Roadmap was cited properly. The reviewer does not have time to check all the references for the authors.

3. In order to improve the readability, instead of writing the mutagenesis primer sequences in the text (Ln455 to 463), the reviewer suggests the authors to list them in a table.

Reviewer #2: It was not clear what significance is provided by the in-silico docking analysis. Also it is only referred to as GRID, a non-expert introduction to what this means and how it works would be useful.

I found the description and nomenclature of the engineered viruses was difficult to follow.

Reviewer #3: See above

PLOS authors have the option to publish the peer review history of their article (what does this mean?). If published, this will include your full peer review and any attached files.

Reviewer #1: No

Reviewer #2: No

Reviewer #3: No
---

## [Editor Report · Decision Letter 1]

15 Jun 2020

Dear Dr. Maree,

Thank you very much for submitting your manuscript "Symmetrical arrangement of positively charged residues around the 5-fold axes of SAT type foot-and-mouth disease virus enhances cell culture of field viruses" for consideration at PLOS Pathogens. As with all papers reviewed by the journal, your manuscript was reviewed by members of the editorial board and by several independent reviewers. The reviewers appreciated the attention to an important topic. Based on the reviews, we are likely to accept this manuscript for publication, providing that you modify the manuscript according to the review recommendations.

The authors have responded appropriately to most comments by the reviewers. However, Reviewer #2 had some specific questions regarding the novelty of the work but more specifically the way the authors referred to prior work and discussed this in the context of this manuscript; The comments in questions are:

Reviewer #2: This is a meticulous study by well-respected authors which has generated robust data which adds some new information to the process of culture adaptation of FMDV. However, the main claims of the study are already very well established. References (cited in the manuscript) 21, 44, 45 and 23, all describe previous culture adaptation of FMDV involving acquisition of positive charge around the five-fold axis. Therefore, the novelty of the current study is not what would normally be consistent with publication in PLoS Pathogens.

What is of more concern is that the introduction and discussion does not make this situation clear. Despite the references being cited within the paper, there is no explanation in the introduction, of these previous studies (e.g. references 21, 44, 45, 23) having already identified acquisition of positive charge around the 5-fold as a mechanism of culture adaptation. This seems bizarre. Instead the following statement is included:

“The genetic alterations associated with increased cell killing activity during cytolytic passages of the SAT serotype FMD viruses in BHK-21 cells are largely unknown. Nevertheless, it has been noted that amino acid substitutions accumulate in their capsids during serial passaging (44–46)”

One of the references at the end of this statement is even from one of the authors of the current study and specifically describes the acquisition of positive charge around the five-fold as a mechanism of culture adaptation of SAT type viruses. In the context of the novel claims of the current manuscript this statement and other omissions appear misleading. Perhaps hopefully all of this is the result of lack of oversight, but I’m afraid it is rather disappointing.

The authors have removed the statement that is mentioned but they have not addressed the central concern of the reviewer. They need to be clear on the response about these concerns.

Sincerely,

Richard J. Kuhn, PhD

Associate Editor

PLOS Pathogens

Michael Diamond

Section Editor

PLOS Pathogens

Kasturi Haldar

Editor-in-Chief

PLOS Pathogens

orcid.org/0000-0001-5065-158X

Michael Malim

Editor-in-Chief

PLOS Pathogens

orcid.org/0000-0002-7699-2064

The authors have responded appropriately to most comments by the reviewers. However, Reviewer #2 had some specific questions regarding the novelty of the work but more specifically the way the authors referred to prior work and discussed this in the context of this manuscript; The comments in questions are:

Reviewer #2: This is a meticulous study by well-respected authors which has generated robust data which adds some new information to the process of culture adaptation of FMDV. However, the main claims of the study are already very well established. References (cited in the manuscript) 21, 44, 45 and 23, all describe previous culture adaptation of FMDV involving acquisition of positive charge around the five-fold axis. Therefore, the novelty of the current study is not what would normally be consistent with publication in PLoS Pathogens.

What is of more concern is that the introduction and discussion does not make this situation clear. Despite the references being cited within the paper, there is no explanation in the introduction, of these previous studies (e.g. references 21, 44, 45, 23) having already identified acquisition of positive charge around the 5-fold as a mechanism of culture adaptation. This seems bizarre. Instead the following statement is included:

“The genetic alterations associated with increased cell killing activity during cytolytic passages of the SAT serotype FMD viruses in BHK-21 cells are largely unknown. Nevertheless, it has been noted that amino acid substitutions accumulate in their capsids during serial passaging (44–46)”

One of the references at the end of this statement is even from one of the authors of the current study and specifically describes the acquisition of positive charge around the five-fold as a mechanism of culture adaptation of SAT type viruses. In the context of the novel claims of the current manuscript this statement and other omissions appear misleading. Perhaps hopefully all of this is the result of lack of oversight, but I’m afraid it is rather disappointing.

The authors have removed the statement that is mentioned but they have not addressed the central concern of the reviewer. They need to be clear on the response about these concerns.
---

## [Editor Report · Decision Letter 2]

22 Jul 2020

Dear Dr. Maree,

We are pleased to inform you that your manuscript 'Symmetrical arrangement of positively charged residues around the 5-fold axes of SAT type foot-and-mouth disease virus enhances cell culture of field viruses' has been provisionally accepted for publication in PLOS Pathogens.

Best regards,

Richard J. Kuhn, PhD

Associate Editor

PLOS Pathogens

Michael Diamond

Section Editor

PLOS Pathogens

Kasturi Haldar

Editor-in-Chief

PLOS Pathogens

orcid.org/0000-0001-5065-158X

Michael Malim

Editor-in-Chief

PLOS Pathogens

orcid.org/0000-0002-7699-2064

The authors have addressed the remaining concerns with this revision. It now is an important contribution to the FMDV field.
---

## [Editor Report · Acceptance letter]

18 Sep 2020

Dear Dr. Maree,

We are delighted to inform you that your manuscript, "Symmetrical arrangement of positively charged residues around the 5-fold axes of SAT type foot-and-mouth disease virus enhances cell culture of field viruses," has been formally accepted for publication in PLOS Pathogens.

Best regards,

Kasturi Haldar

Editor-in-Chief

PLOS Pathogens

orcid.org/0000-0001-5065-158X

Michael Malim

Editor-in-Chief

PLOS Pathogens

orcid.org/0000-0002-7699-2064